# Evaluation of the Anti-Obesity Effect of Zeaxanthin and Exercise in HFD-Induced Obese Rats

**DOI:** 10.3390/nu14234944

**Published:** 2022-11-22

**Authors:** Mona Al-thepyani, Salha Algarni, Hana Gashlan, Mohamed Elzubier, Lina Baz

**Affiliations:** 1Department of Biochemistry, Faculty of Science, King AbdulAziz University, Jeddah 21589, Saudi Arabia; 2Department of Chemistry, College of Sciences & Arts, King Abdulaziz University, Rabigh 21911, Saudi Arabia; 3Biochemistry Department, Faculty of Medicine, Umm Al-Qura University, Al Abdeyah, Makkah P.O. Box 7607, Saudi Arabia

**Keywords:** zeaxanthin, exercise, swimming, anti-obesity, high-fat diet, antioxidants, adipokines, anti-inflammatory, lipid profiles

## Abstract

Obesity is a worldwide epidemic associated with many health problems. One of the new trends in health care is the emphasis on regular exercise and a healthy diet. Zeaxanthin (Zea) is a carotenoid with many beneficial effects on human health. The aim of this study was to investigate whether the combination of Zea and exercise had therapeutic effects on obesity induced by an HFD in rats. Sixty male Wistar rats were randomly divided into five groups of twelve: rats fed a standard diet; rats fed a high-fat diet (HFD); rats fed an HFD with Zea; rats fed an HFD with Exc; and rats fed an HFD with both Zea and Exc. To induce obesity, rats were fed an HFD for twelve weeks. Then, Zea and exercise were introduced with the HFD for five weeks. The results showed that the HFD significantly increased visceral adipose tissue, oxidative stress, and inflammation biomarkers and reduced insulin, high-density lipoprotein, and antioxidant parameters. Treatments with Zea, Exc, and Zea plus Exc reduced body weight gain, triacylglycerol, glucose, total cholesterol, and nitric oxide levels and significantly increased catalase and insulin compared with the HFD group. This study demonstrated that Zea administration and Exc performance appeared to effectively alleviate the metabolic alterations induced by an HFD. Furthermore, Zea and Exc together had a better effect than either intervention alone.

## 1. Introduction

Obesity is an alarmingly increasing global public health problem. In addition, it is becoming a significant economic and health burden [1]. Obesity is a complicated disease that is characterized by a significant increase in body fat mass. It is responsible for increasing the risk of developing diseases such as dyslipidemia, type 2 diabetes mellitus, myocardial infarction, coronary heart disease, arterial hypertension, ovarian polycytosis, fatty liver disease, and various cancers; consequently, it results in a shorter life expectancy [2,3]. Obesity results from a multifaceted and complex interaction between environmental factors, human behavior, and genetic susceptibility. Unhealthy lifestyles (such as a sedentary lifestyle and a high-fat diet) are strong predictors of obesity, type 2 diabetes, and insulin resistance [4]. Obesity is a critical factor in the development of metabolic abnormalities in glucose, lipid metabolism, dyslipidemia, oxidative stress, and chronic inflammation [5,6,7,8]. These metabolic abnormalities of obesity may be exacerbated by an abnormal inflammatory response and the increased production of reactive oxygen species. It is well-known that a proinflammatory response is associated with the production of toxic reactive oxygen species and the subsequent generation of oxidative stress. Adipocytes from obese individuals exhibit an altered adipokine profile with increased expression and secretion of proinflammatory cytokines [9].

Due to the undesirable side effects of anti-obesity medications and the numerous risks associated with surgical interventions, the combination of regular exercise and healthy diet remains the safest and most cost-effective method for obesity management [10,11]. Furthermore, because natural products are safe and nontoxic, consuming functional foods is one of the best treatment options for weight loss [12]. In addition, the high safety margins of functional foods, especially when used in long-term regimens, generates interest in their use for managing obesity compared with synthetic medications [10].

Carotenoids are widely used terpenoid pigments. They are endowed with a system of conjugated double bonds, which yield their colors. Carotenoids are categorized into oxygen-free carotenes and their oxygenated derivatives, the xanthophylls. Furthermore, they are differentiated according to the structures of their end groups (acyclic or linear, monocyclic, and bicyclic) [13]. Zeaxanthin (Zea), from the xanthophyll family, is a carotenoid pigment that can be found in a wide range of fruits and vegetables and is more abundant in eggs, yellow corn, orange bell pepper, and green leafy vegetables [14,15,16]. Zea has also been proven to exert a variety of beneficial effects on human health due to its capability to scavenge free radicals, exhibit antioxidant effects, and reduce inflammation [16]. The oral administration of Zea to obese mice has improved dyslipidemia and decreased the progression of obesity [17]. Tuzcu et al. found that rats fed an HFD showed increased oxidative stress and the upregulation of inflammatory markers. However, the administration of Zea significantly reduced oxidative damage by increasing antioxidant enzyme activities and reducing malondialdehyde (MDA) concentrations.

Physical exercise (Exc) is an important intervention for treating and preventing various diseases such as osteoarthritis, obesity, Alzheimer’s disease, and hypertension [18]. In addition, regular exercise training improves lipid metabolism and oxidative capacity and lowers blood pressure, insulin resistance, and serum triglycerides [19,20,21].

Recent research on healthcare emphasizes regular exercise and a healthy diet. How diet and exercise training interact to produce synergistic or antagonistic effects on physiological functions that lead to health promotion and the possible mechanisms underlying such interactions are still poorly understood [22]. The combination of functional food consumption and exercise can cause changes in body metabolism [23]. However, there are no data on the therapeutic impact of zeaxanthin supplementation in combination with exercise for weight loss or to reduce the problems associated with obesity. This study investigated the effects of Zea, Exc, and the combination of Zea and Exc on obesity induced by an HFD in male Wistar rats.

## 2. Materials and Methods

### 2.1. Animals and Diets

Sixty seven-week-old male Wistar rats weighing (200 ± 10) g were obtained from the animal house unit of King Fahad Medical Research Center (KFMRC), Jeddah, Saudi Arabia. The animals were kept in ambient conditions (22 ± 2 °C, 50% humidity, 12/12 h light/dark) for one week before the initiation of the experiments. A balanced commercial diet and tap water were provided throughout the experiment. The study was approved by the Ethics Committee of King Abdulaziz University (approval No. 518-20) and compiled with the rules and regulations of the Animal Care and Use Committee at KFMRC. A standard chow diet (Grain Silos & Flour Mills Organization, KSA) was obtained from the KFMRC. The standard diet (387 kcal/100 g) contained 67.8% carbohydrates, 20% protein, 4% fat, 3.5% fiber, 0.3% choline chloride, and 4.5% vitamins and minerals. The high-fat diet (HFD) was prepared by mixing the standard diet (59.6%) with commercial butter (29.79%), casein (7.45%), DL-methionine (0.2%), and a vitamin and mineral mix (2.97%) [24]. The HFD (529 kcal/100 g) contained 40.4% carbohydrates, 19.4% protein, 32.2% fat, 2.1% fiber, 0.1% choline chloride, 0.2% methionine, and 5.6% vitamins and minerals. Zeaxanthin (purity of 98%) extracted from marigold flowers was provided by Xi’an Tongze Biotech Co., Ltd., Xi’an, China.

### 2.2. Experimental Design

The experimental design is shown in Figure 1. After a one-week acclimation period, 60 mice were randomly divided into two groups: the normal group was fed a standard diet (control, *n* = 12), and the experimental group was fed a high-fat diet (HFD, *n* = 48). At week 12, after confirming the body weight difference, the 48 obese rats were divided into four groups (*n* = 12/group): group I—positive control group (HFD); group II—HFD with the daily oral administration of 100 mg/kg of zeaxanthin (HFD + Zea); group III—HFD with exercise (HFD + Exc); and group IV—HFD with the daily oral administration of 100 mg/kg of zeaxanthin combined with exercise (HFD + Zea/Exc). All obese groups were continually fed the HFD during the experimental period (17 weeks). Sunflower oil was used as a vehicle; all groups received the same volume of sunflower oil (~2 mL/kg/day) [25,26]. A dosage of 100 mg/kg of zeaxanthin was selected because many in vivo studies have shown that it is an antioxidant at this dosage [27,28,29]. At the end of the experiment, rats fasted overnight and were anesthetized with isoflurane (induced with 5% isoflurane and maintained with 3% in a 2 L/min oxygen flow in a sealed container). Blood was drained from the retro-orbital veins of the rats, and serum was obtained from the blood by centrifugation at 3000 rpm for 15 min and was stored at −80 °C for biochemical analysis. Liver samples were frozen in liquid nitrogen and stored at −80 °C until analysis.

### 2.3. Exercise Protocol

In this study, swimming was used as a model exercise intervention. Rats in the swimming groups (HFD + Exc and HFD + Zea/Exc) swam in a plastic swimming tank with dimensions of 78 × 56 × 48 cm. The depth of the water was sufficient to preclude resting and prevent bobbing. To avoid hypothermia, the water temperature was set at 32 ± 2 °C. The rats swam for 10 min on the first day. The exercise time was increased by 10 min each day. In the following weeks, the rats were allowed to swim for 60 min once per day for five weeks [30,31,32].

### 2.4. Measurement of Body Weight, Food Intake, and Relative Weight of Visceral Adipose Tissue

During the experiment, the body weights of all experimental rats were recorded at the beginning of the experiment (time 0) and then weekly. The weight gain of the rats was also monitored weekly. The food intake (difference between the food offered and the food remaining) was measured daily, and the energy intake was calculated by multiplying the food intake by the total energy of the standard diet and the HFD [33]. At the end of the study, the weights of the visceral adipose tissues of the control and experimental groups were determined and noted. The ratio of visceral adipose tissue to body weight was expressed as the relative weight per 100 g of body weight [34]. The following formula was used to calculate the relative visceral adipose tissue:$$\mathrm{The}\;\mathrm{ratio}\;\mathrm{of}\;\mathrm{visceral}\;\mathrm{fat}\;\mathrm{to}\;\mathrm{body}\;\mathrm{weight}=\frac{\;\mathrm{visceral}\;\mathrm{adipose}\;\mathrm{tissue}\;(g)}{\;\mathrm{final}\;\mathrm{body}\;\mathrm{weight}\;(g)}\;\times \;100$$

### 2.5. Determination of Serum Biochemical Parameters

The levels of serum glucose, triacylglycerol (TGA), total cholesterol (TC), and high-density lipoprotein (HDL) were estimated using specific standard diagnostic kits with an automated modular analyzer (COBAS^®^ 8000 series) at Alnoor Specialist Hospital, Makkah, Saudi Arabia. The levels of very low density lipoprotein (VLDL) and low-density lipoprotein (LDL) were calculated based on Friedewald’s equation [35]:$$\mathrm{LDL}=\mathrm{TC}-\mathrm{HDL}-\left(\frac{\mathrm{TGA}}{5}\right)$$
VLDL=TGA/5

Serum insulin and the inflammatory markers leptin, resistin, and adiponectin were determined in serum using enzyme-linked immunosorbent assay (ELISA) kits (Solarbio Science & Technology, Beijing, China). The oxidative stress markers malondialdehyde (MDA) and nitric oxide (NO) as well as the activities of the antioxidant enzymes catalase (CAT) and superoxide dismutase (SOD) were evaluated in hepatic tissue using colorimetric assay kits from Solarbio Science & Technology (Beijing, China).

### 2.6. Histological Analysis of Visceral Adipose Tissue

Visceral adipose tissues were fixed in 10% buffered formalin, embedded in paraffin, and sectioned into 4 μm thick sections. These sections were stained with hematoxylin and eosin (H & E), and histological images were examined through a light microscope (Olympus BX51) under 20× magnification. Adipocyte size and diameter were calculated using ImageJ version 1.53t (National Institutes of Health).

### 2.7. Statistical Analysis

The data were analyzed using GraphPad Prism 8 (GraphPad Inc., La Jolla, CA, USA). Data are shown as the means ± standard errors of the means (means ± SEM). A one-way ANOVA was used to assess the significant differences between different groups, followed by Tukey’s post hoc test. Differences with *p* < 0.05 were considered statistically significant.

## 3. Results

### 3.1. Effects of Zea and Exc on Body Weight, Weight Gain, Food Intake, and Relative Visceral Adipose Tissue in HFD-Induced Obese Rats

As shown in Table 1, at the end of 12 weeks of obesity induction, rats fed an HFD showed a significant increase in body weight (*p* < 0.01) compared with the rats fed a standard diet (Figure 2). After five weeks of treatment, the final body weight and weight gain had decreased, but not statistically significantly, compared with the HFD group. Moreover, groups that underwent Exc (HFD + Exc and HFD + Zea/Ex) showed decreases in their body weights compared with HFD + Zea, but no significant difference was observed. Food and energy intake were significantly increased in the control group compared with the HFD-fed groups due to the low calorie content in the standard diet. Additionally, the HFD groups had no significant differences in food and energy intake. Furthermore, the relative weights of visceral adipose tissue were increased in the HFD group and slightly decreased in the treated groups (Table 1).

### 3.2. Effects of Zea and Exc on Blood Glucose and Insulin Levels

The determination of blood glucose and insulin levels in control and obese rats is shown in Figure 3. There was a highly significant increase (*p* < 0.001) in blood glucose levels in the HFD group compared with the control group. On the other hand, obese rats treated with Zea and/or Exc showed a significant reduction in glucose levels compared with the HFD group (Figure 3A). In addition, serum insulin levels were decreased in the HFD group compared with the control group, with no significant differences. Interestingly, all treated groups (HFD + Zea, HFD + Exc, and HFD + Zea/Exc) exhibited significantly elevated (*p* < 0.05) insulin levels compared with the HFD group (Figure 3B).

### 3.3. Effects of Zea and Exc on Serum Lipid Profiles

The effects of Zea and Exc on the serum lipids of HFD-fed rats are illustrated in Figure 4. It was found that the HDL levels in the HFD group were lower than in the control group but did not differ significantly. Notably, the HDL levels among the treated groups (HFD + Zea, HFD + Exc, and HFD + Zea/Exc) were increased compared with the HFD group and were statistically elevated (*p* < 0.01) in the Exc groups (HFD + Exc and HFD + Zea/Exc) (Figure 4A). In addition, the group that did not perform Exc (HFD + Zea) had significantly lower (*p* < 0.05) HDL levels than the group that underwent Exc.

The LDL levels in the HFD group were higher than in the control group, without significant differences. In contrast, all the treated groups had low LDL levels compared with the HFD group, especially the Exc groups (HFD + Exc and HFD + Zea/Exc), which were significantly decreased (*p* < 0.05) compared with the HFD group (Figure 4B). Interestingly, the HFD + Zea/Exc group had lower LDL levels than HFD + Exc alone, and there was no significant difference.

Furthermore, the TC levels in the HFD group were significantly elevated (*p* < 0.05) compared with the control group, and the TC levels were slightly reduced in all treated groups, without significant differences (Figure 4C). Finally, the TAG and VLDL levels showed highly significant increases in the HFD group compared with the control group (*p* < 0.001). On the other hand, the HFD + Zea and HFD + Exc groups had low levels of TAG and VLDL compared with the HFD group, especially HFD + Zea, which was significantly decreased (*p* < 0.05). In addition, the levels of TGA and VLDL in the HFD + Zea group were significantly decreased (*p* < 0.05) compared with the HFD + Zea/Exc group, which was similar to the HFD group (Figure 4D,E).

### 3.4. Effects of Zea and Exc on Serum Inflammatory Markers

As illustrated in Figure 5, there was a significant increase (*p* < 0.01) in the leptin levels in the HFD group compared with the control group. Furthermore, the groups treated with Zea exhibited low levels of leptin compared with the HFD group, especially HFD + Zea/Exc, which was significantly decreased (*p* < 0.05) compared with the HFD group and exercise alone (Figure 5A). Meanwhile, the resistin levels were significantly increased (*p* < 0.001) in the HFD group compared with the control group. Additionally, the resistin levels in the HFD + Zea and HFD + Zea/Exc groups were significantly reduced (*p* < 0.01). However, there was no statistical decrease in the HFD + Exc group compared with the HFD group (Figure 5B). On the other hand, the adiponectin levels were reduced in the HFD group compared with the control group, with no significant difference. Nevertheless, increased adiponectin levels were observed in all treated groups compared with the HFD group (Figure 5).

### 3.5. Effects of Zea and Exc on Oxidative Stress and Antioxidant Markers in Liver Tissue

The oxidative stress and antioxidant markers were measured in hepatic tissue (Figure 6). CAT activity was significantly reduced (*p* < 0.05) in the HFD group compared with the control group. Moreover, the levels of CAT in the HFD + Zea, HFD + Exc, and HFD + Zea/Exc treated groups were significantly increased (*p* < 0.001) compared with the HFD group (Figure 6A). Additionally, the SOD activity was lower than in the control group, with no significant difference. Intriguingly, groups receiving Zea (HFD + Zea and HFD + Zea/Exc) exhibited significant increases in SOD levels (*p* < 0.01) compared with the HFD group and the HFD + Exc group (Figure 6B).

In the current study, the hepatic MDA levels were significantly increased (*p* < 0.001) in the HFD group compared with the control group. However, after intervention with Zea and Exc, the level of MDA was significantly reduced (*p* < 0.001) compared with the HFD group. Additionally, it was noted that groups treated with Zea (HFD + Zea and HFD + Zea/Exc) exhibited lower levels of MDA compared with the HFD + Exc group, with no significant difference (Figure 6C). No significant increase was observed in the levels of nitric oxide (NO) in the HFD group in comparison with the control group. Additionally, all treated groups had reduced NO levels compared with the HFD group (Figure 6D).

### 3.6. Effects of Zea and Exc on Visceral Adipose Tissue

As shown in Figure 7A, the HFD group exhibited an accumulation of visceral adipose tissue compared with the control group. However, the rats treated with either Zea or Exc showed no significant decrease in visceral adipose tissue accumulation compared with the HFD control group. The degree of lipid accumulation in the white adipose tissue is proportional to the size of the tissue [36,37]. The histological analysis of visceral adipose tissue showed that adipocytes were enlarged in the HFD group compared with the control group, and this effect was visible with the naked eye. In addition, the groups administered Zea and Exc (HFD + Zea, HFD + Exc, and HFD + Zea/Exc) showed smaller adipocytes and similar histology to the control group (Figure 7B). The sizes and diameters of the adipocytes in visceral adipose tissues were significantly increased (*p*< 0.001) in the HFD group compared with the control group. In contrast, it was found that the groups treated with Zea and Exc (HFD + Zea, HFD + Exc, and HFD + Zea/Exc) had significantly decreased (*p* < 0.001) adipocyte sizes and diameters compared with the HFD group (Figure 7C,D). Notably, there were no differences in the sizes and diameters of adipocytes between treated groups.

## 4. Discussion

Obesity is a pathological problem caused by an unhealthy diet and lack of exercise. A healthy low-fat diet, more physical activity, and behavioral changes are all part of the current best practices for treating obesity [38]. Recently, there has been extensive research on using natural products to treat obesity and reduce its complications instead of medical drugs with unwanted side effects. Additionally, physical exercise can reduce the adverse effects of obesity, even if weight loss does not occur [39]. In this study, the anti-obesity effects of Zea and Exc in HFD-induced obese rats were investigated.

HFD-induced obesity in rodents is considered a suitable model to study the potential roles of dietary fat in the incidence and progression of obesity because they are reportedly very similar to human obesity [40,41]. In this study, the induction of obesity by feeding rats an HFD was successful after 12 weeks, consistent with a previous study [42]. All rats fed an HFD during the 12 weeks of induction had significantly increased body weight compared with the control group. However, after treatment with Zea and/or Exc from the end of the 12th week to the 17th week of the experiment, rats in the HFD + Zea, HFD + Exc, and HFD + Zea/Exc groups had slower rates of weight gain, lower body weights, and less fat mass. The results of the current study suggest that Zea and Exc inhibit HFD-induced obesity by preventing increased body weight and visceral fat. According to a previous study [17], treatment with Zea for four weeks significantly decreased body weight, weight gain, and adipose mass in HFD-induced obese mice. Additionally, the groups that underwent Exc (HFD + Exc and HFD + Zea/Exc) exhibited lower body mass and weight gain than those only treated with Zea (HFD + Zea), which suggests that physical exercise can prevent the progression of obesity or at least mitigate it. This result agrees with previous studies reporting that physical activity and regular exercise can increase energy metabolism in the body, helping to control body weight and reduce obesity [43,44].

The reduction in body weight reflects the negative status of energy expenditure, which could result from a reduction in food intake or an increase in energy expenditure [45]. In our study, the Exc groups (HFD +Exc and HFD + Zea/Exc) showed less feed intake and subsequently reduced caloric intake; however, no significant difference was observed when compared with the HFD group. Similarly, other studies found that feeding rats a high-fat diet causes them to feed less and consume fewer calories when performing physical exercise [46,47,48]. One reason for this might be that physical exercise, which is a stress agent, may affect the production of anorectic hormones and increase the sensitivity of their receptors, thereby suppressing the sensation of hunger and controlling energy balance during physical exercise [49,50]. Moreover, other studies have shown that physical activity can reduce food intake and increase energy expenditure, which was also the case in this study [51,52]. Several studies found that regular exercise can improve some of the metabolic complications of obesity, even in the absence of weight loss [39].

Obesity is associated with over 200 health complications, such as type 2 diabetes mellitus and insulin resistance, and it is the fifth largest cause of mortality globally [53]. Previous reports have shown that blood glucose levels were significantly elevated in rats fed an HFD compared with a normal healthy group [54,55].

In the current study, the HFD group exhibited significantly higher blood glucose levels than the control group. However, the supplementation of Zea and/or the induction of Exc led to a significant decrease in blood glucose levels in the HFD + Zea, HFD + Exc, and HFD + Zea/Exc groups. This finding agrees with Tuzcu et al., who showed that Zea reduced glucose levels in rats fed an HFD [28]. Zea has also been proven to lower blood sugar levels, relieve diabetes symptoms, and improve cognitive deficits in diabetic animals [56,57]. In addition, exercise training has been reported to improve glucose levels, insulin sensitivity, and pancreatic β-cell function [58,59,60]. Similar to the current findings, previous studies have demonstrated that glucose uptake efficiency improves with exercise and is closely related to physical load and exercise intensity. Furthermore, Exc increases glucose absorption by stimulating insulin and regulating the response of cells to insulin [61,62]. On the other hand, this study found that insulin levels had decreased in the HFD group and increased in the control and treated groups. This result is in contrast to what was expected and what was demonstrated in previous studies. A possible explanation for these results could be a defect in pancreatic β-cells leading to a decrease in insulin production. According to previous research by Gupta et al., the function and viability of β cells depend on insulin signaling; obesity-related abnormalities (lipotoxicity, glucotoxicity, increased oxidative stress, and inflammation) are known to lead to impaired insulin secretion and hyperglycemia [63]. However, it is imperative to point out that the proper diagnosis of insulin resistance would require performing glucose tolerance tests (GTT) and insulin tolerance tests (ITT). Therefore, these findings lay a foundation for future investigations focusing on diabetes as a complication of obesity using HFD-induced obese rats as a diabetic model.

Serum lipid profiles are good predictors of the development of diseases related to metabolic disorders caused by obesity [64]. According to previous studies, an HFD has been associated with dyslipidemia, which is characterized by elevated total LDL, VLDL, TC, and TGA levels and decreased HDL [65,66]. In this study, rats fed an HFD had high levels of LDL, VLDL, TC, and TGA and low HDL levels, suggesting the development of dyslipidemia. In contrast, we found that the LDL, VLDL, TC, and TGA levels were reduced in the HFD + Zea and HFD + Exc groups, whereas HDL levels were increased, which is consistent with the findings of previous studies [28,67]. Furthermore, compared with the HFD and other treated groups, the combination of Zea and Exc had the highest HDL levels, whereas the LDL levels were significantly decreased. Moreover, the levels of TC in the combination group were low, as in the other treated group, without significant difference. However, in contrast to the HFD + Zea and HFD + Exc groups, the TGA levels in the HFD + Zea/Exc group were similar to the levels in the HFD group, in agreement with previous studies [17,21]. Our results demonstrate that Zea supplementation might prevent dyslipidemia and improve lipid profiles in HFD-fed rats. Furthermore, as one of the carotenoids, Zea might stimulate several nuclear receptors that regulate the transcription of several lipid metabolism pathways [68]. On the other hand, HFD-induced lipid metabolism disorders may be recovered with a suitable exercise intervention, which agrees with previously published research [44,69,70]. Mika et al. reported that exercise increases lipolysis, reduces the uptake of fatty acids by adipocytes, alters the composition of fatty acids in adipose tissue, and modifies the expression of the enzymes responsible for the synthesis, elongation, and desaturation of fatty acids [71]. Exercise also encourages the “beiging” of adipose tissue and helps to increase mitochondrial activity, which leads to a rise in fatty acid oxidation in the adipose tissue [71].

Obesity is characterized by alterations in circulating adipokine concentrations due to the excessive accumulation and dysfunction of adipose tissue [72]. Leptin, resistin, and adiponectin are major adipokines secreted from white adipose tissue, and they play an essential role in regulating metabolism [73,74]. Increased circulating leptin and resistin levels with decreased adiponectin levels are characteristics of obesity [75,76]. Leptin and resistin are proinflammatory adipokines associated with energy homeostasis, the central control of food intake, inflammation, cardiovascular disease, and insulin resistance [77,78,79], while adiponectin has antidiabetic, anti-inflammatory, and antiatherogenic properties [80]. In the current study, rats fed an HFD exhibited significantly increased leptin and resistin levels with a parallel decrease in adiponectin, which is in agreement with prior findings [81]. On the other hand, leptin levels decreased in groups that were provided Zea (HFD + Zea and HFD + Zea/Exc), especially the HFD + Zea/Exc group, which exhibited significantly reduced leptin levels compared with Exc alone. Zea and Exc significantly reduced resistin levels in all treated groups, particularly in the HFD + Zea and HFD + Zea/Exc groups. Additionally, adiponectin was increased in all treated groups compared with the HFD groups, with no significant difference. It has been reported that leptin, resistin, and adiponectin, which are involved in mediating obesity, have been linked to increased oxidative stress and inflammation, which may cause a global health risk leading to hyperglycemia, cancer, diabetes, and other diseases affecting vital organs [78,82]. Thus, from this point, we can explain the effect of Zea on leptin, resistin, and adiponectin levels by its activity as an antioxidant and anti-inflammatory agent, which agrees with previous studies [16,83]. On the other hand, the role of exercise in lowering leptin and resistin and increasing adiponectin can be explained by two reasons. First, physical exercise not only contributes to a reduction in adipose tissue mass but may also attenuate the inflammation caused by obesity through the dysregulated expression of adipokines secreted by adipose tissue [84]. Second, during exercise, muscles produce myokines and release them into the circulation, which may balance and counteract the harmful effects of proinflammatory adipokines [84,85,86,87].

Obesity and its associated disorders are closely associated with oxidative stress [88], which is defined as an imbalance between the oxidative and antioxidant systems that results in impaired redox signaling [89,90]. In addition, it is known that HFD-induced obesity leads to oxidative damage through lipid peroxidation by producing malondialdehyde, depleting endogenous antioxidants, and decreasing the activity of antioxidant enzymes such as SOD and CAT [91,92,93]. Moreover, nitric oxide and other oxidative stress markers appear to be important factors in the association between obesity and other diseases [94]. In the present study, after five weeks of administration of Zea and/or Exc performance, the levels of CAT and SOD were significantly increased, while the MDA levels were significantly decreased compared with the HFD group. Additionally, the groups treated with Zea (HFD + Zea and HFD + Zea/Exc) had the highest levels of SOD when compared with the HFD group. In addition, the NO levels were lower in all treated groups. From the current results, it appears that both Zea and Exc may improve oxidative stress by increasing the activity of CAT and SOD and reducing MDA and NO, which resulted from feeding the rats an HFD. However, it is worth noting that the effect of Zea was slightly greater than the effect of exercise (Figure 7). According to the findings of previous studies [28,95,96], Zea may exert its antioxidant properties by directly suppressing reactive oxygen species. Similar to other carotenoids, Zea has a chain of isoprene residues with conjugated double bonds, which allows it to accept an extra electron and absorb energy from excited molecules [96]. Additionally, the potential role of exercise in suppressing oxidative stress is notable. It has previously been reported that after exercise the body produces more endogenous antioxidants, which help protect cells from oxidative damage [97].

Obesity is initiated by the accumulation of free fatty acids in adipose tissue, which can lead to enlarged adipocytes [36,37]. In the present study, histological and statistical analyses of visceral adipose tissue showed that the sizes and diameters of adipocytes were smaller in rats treated with Zea and Exc (HFD + Zea, HFD + Exc, and HFD + Zea/Exc groups) compared with the HFD group. This is consistent with a previous study [17] in which the sizes of larger adipocytes were significantly reduced after four weeks of treatment with Zea. Furthermore, regular exercise training has been shown to reduce adipocyte hypertrophy in both subcutaneous and visceral adipose tissue [98,99,100], which is consistent with our findings in this study.

To the best of our knowledge, there was no previous study on the combination of zeaxanthin and exercise; thus, we can infer that the combination of zeaxanthin and exercise has greater efficacy in reversing the complications of obesity (Figure 8).

## 5. Conclusions

This study showed that the induction of obesity by feeding an HFD to rats was responsible for significant weight gain, increased energy intake, and visceral adipose tissue accumulation. Moreover, HFD-induced obesity was associated with metabolic disorders. This resulted in abnormal values of most biochemical parameters that were studied, such as glucose, insulin, and lipid profiles, and disturbances in the inflammatory and oxidative stress parameters. After weeks of treatment, we found that zeaxanthin supplementation and exercise exerted an anti-obesity effect by reducing visceral fat mass, effectively improving dyslipidemia, and lowering blood glucose levels. Moreover, these interventions were able to suppress oxidative stress by increasing the activities of antioxidant enzymes and depleting ROS. Additionally, Zea and Exc decreased proinflammatory adipokines. In conclusion, the current study demonstrates that zeaxanthin and exercise have beneficial effects in treating and preventing obesity and its complications. Finally, zeaxanthin and exercise together had a better effect than either intervention alone. Future studies with a more extended treatment duration of more than five weeks are recommended. Further research could be conducted to determine the effect of Zea on other HFD complications, such as diabetes, cardiovascular disease, and renal dysfunction. In addition, it is essential to perform further studies on other types of exercise with adequate intensity and appropriate durations.

## Figures and Tables

**Figure 1 nutrients-14-04944-f001:**
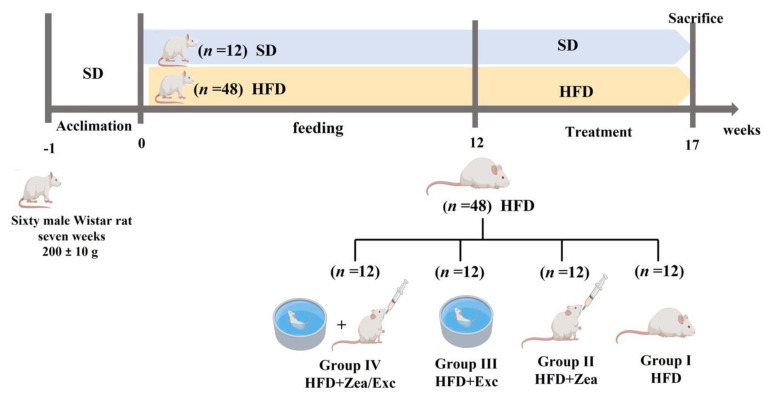
Experimental design.

**Figure 2 nutrients-14-04944-f002:**
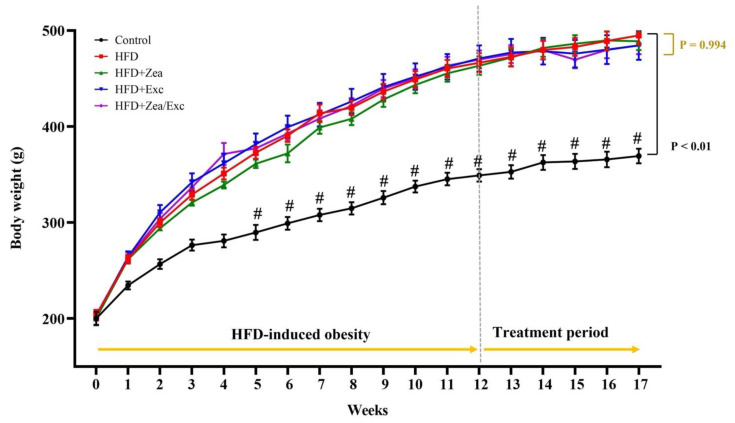
Effects of Zea and Exc on body weight in HFD-induced obese rats. Data are expressed as means ± SEM, *n* = 12. # *p* < 0.05 vs. control.

**Figure 3 nutrients-14-04944-f003:**
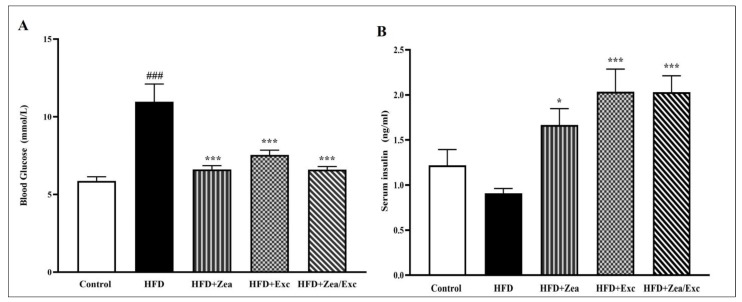
Effects of Zea and Exc on blood glucose levels (**A**) and insulin levels (**B**) in HFD-induced obese rats. Data are expressed as means ± SEM, *n* = 7. ### *p* < 0.001 vs. control; * *p* < 0.05, *** *p* < 0.001 vs. HFD.

**Figure 4 nutrients-14-04944-f004:**
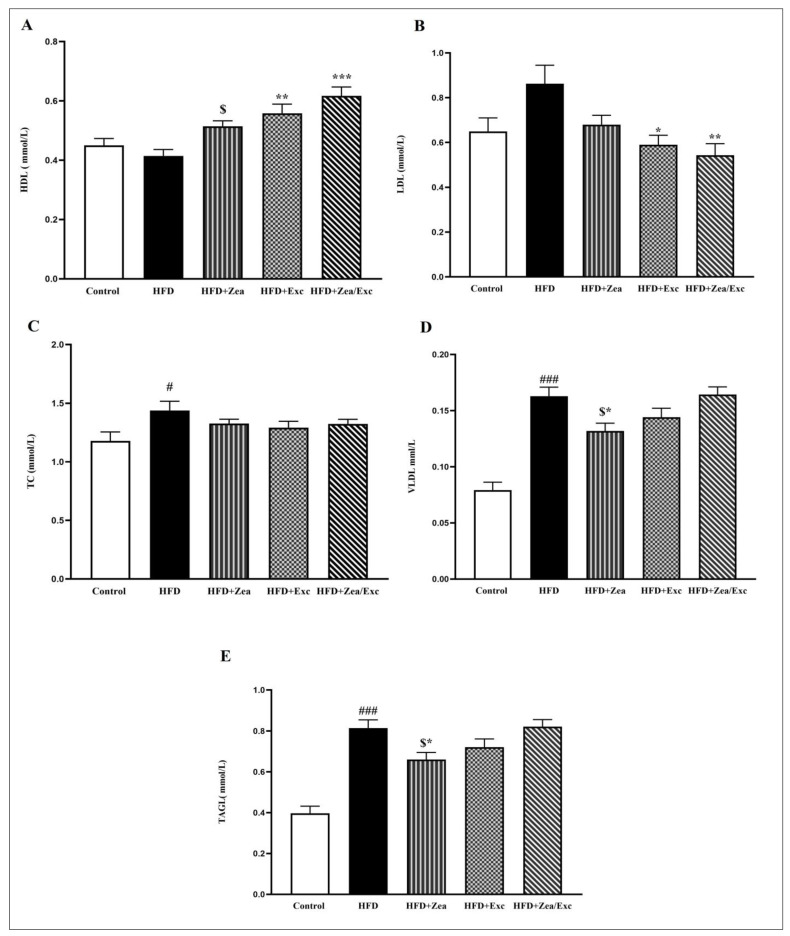
Effects of Zea and Exc on the serum lipid profiles in HFD-induced obese rats. (**A**) High-density lipoprotein (HDL); (**B**) low-density lipoprotein (LDL); (**C**) total cholesterol (TC); (**D**) very low density lipoprotein (VLDL); (**E**) triacylglycerol (TAG). Data are expressed as means ± SEM, *n* = 12. # *p* < 0.05, ### *p* < 0.001 vs. control; * *p* < 0.05, ** *p* < 0.01, *** *p* < 0.001 vs. HFD; $ *p* < 0.05 vs. HFD + Zea/Ex.

**Figure 5 nutrients-14-04944-f005:**
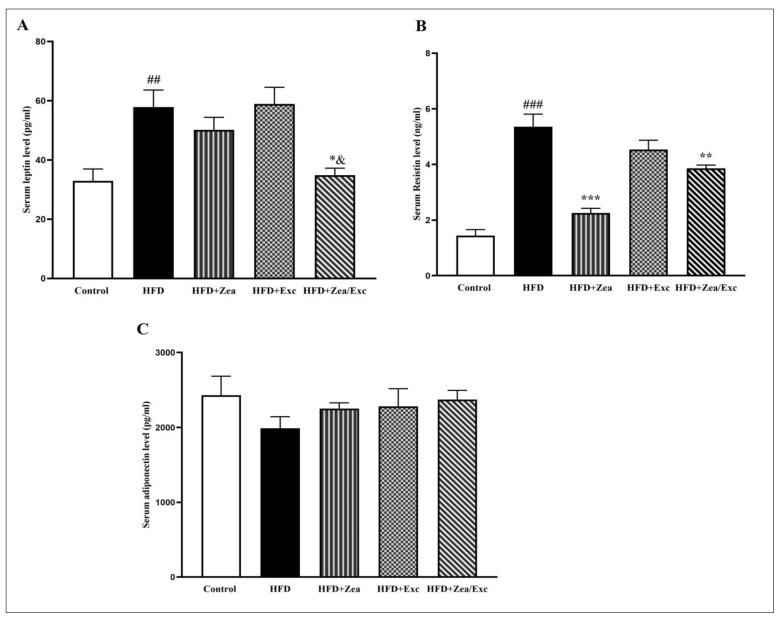
Effects of Zea and Exc on serum leptin levels (**A**), serum resistin levels (**B**), and adiponectin levels (**C**) in HFD-induced obese rats. Data are expressed as means ± SEM, *n* = 7. ## *p* < 0.01, ### *p* < 0.001 vs. control; * *p* < 0.05, ** *p* < 0.01, *** *p*≤ 0.001 vs. HFD; & *p* ≤ 0.05 vs. HFD + Ex.

**Figure 6 nutrients-14-04944-f006:**
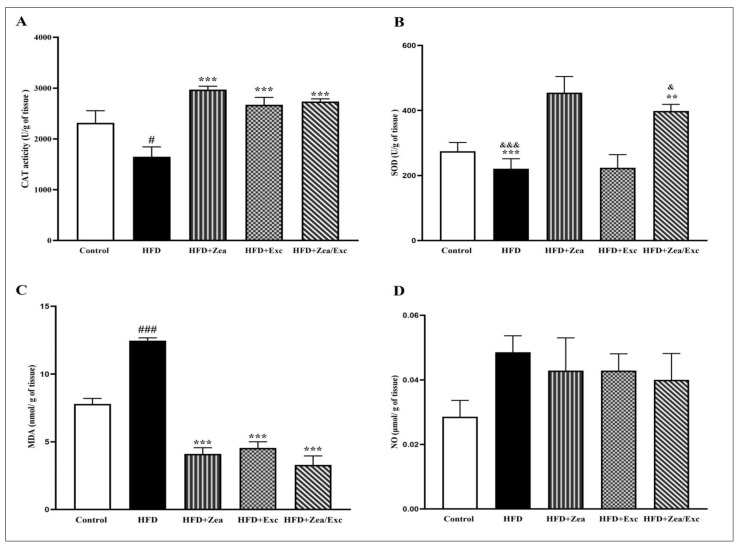
Effects of Zea and Exc on hepatic tissue oxidative stress and antioxidant markers in HFD-induced obese rats. (**A**) Catalase (CAT); (**B**) superoxide dismutase (SOD); (**C**) malondialdehyde (MDA); (**D**) nitric oxide (NO). Data are expressed as means ± SEM, *n* = 7. # *p* < 0.05, ### *p*< 0.001 vs. control; ** *p* <0.01, *** *p* <0.001 vs. HFD; & *p* < 0.05, &&& *p* < 0.001 vs. HFD + Exc.

**Figure 7 nutrients-14-04944-f007:**
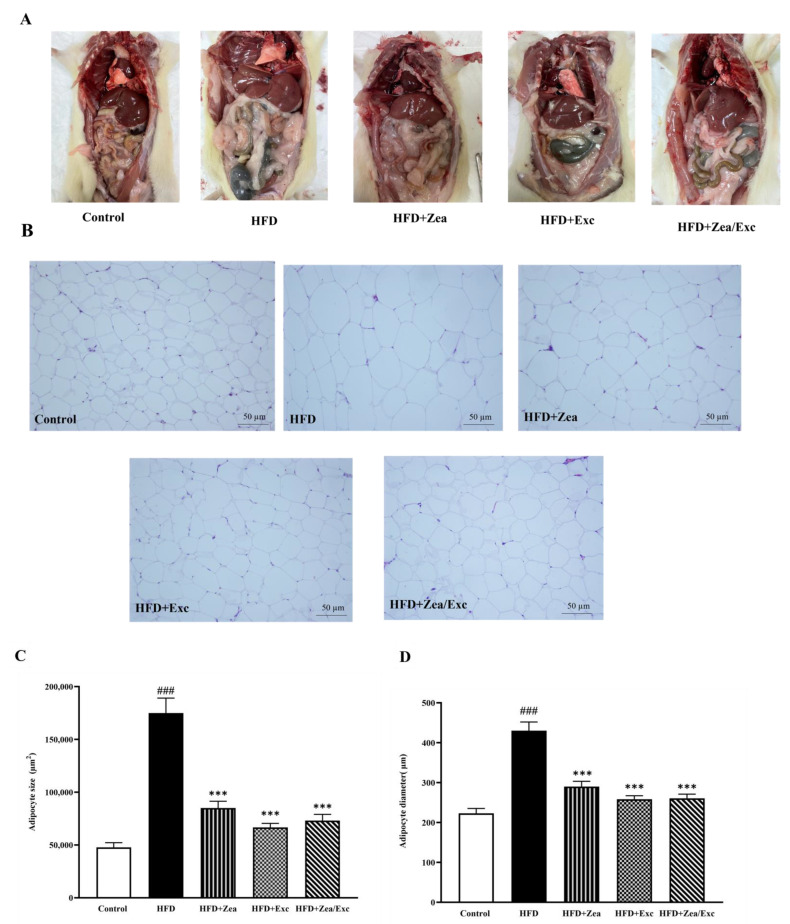
Effects of Zea and Exc on visceral adipose tissue in HFD-induced obese rats. (**A**) Abdominal cavity dissection of visceral adipose tissue; (**B**) histology of visceral adipose tissue; (**C**) adipocyte size; (**D**) adipocyte diameter. Data are expressed as means ± SEM, *n* = 4. ### *p* < 0.001 vs. control; *** *p* < 0.001 vs. HFD; 20× magnification.

**Figure 8 nutrients-14-04944-f008:**
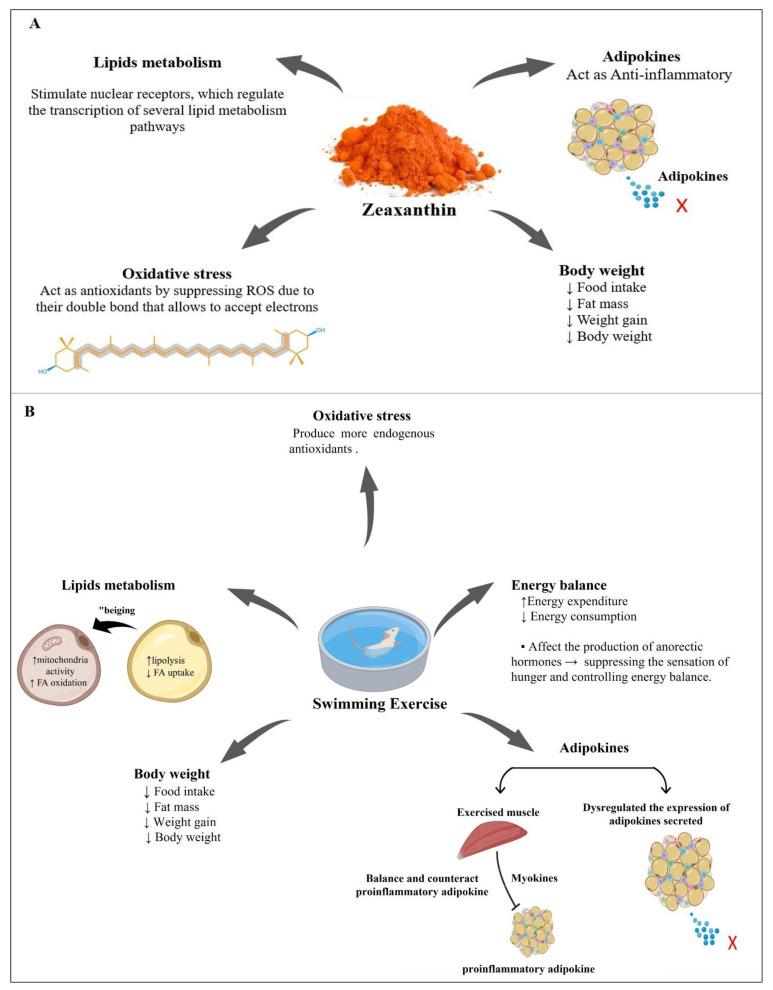
Possible mechanisms of zeaxanthin (**A**) and exercise (**B**) leading to the reduction of the complications of obesity and contributing to its treatment. ↑ = increase, ↓ = decrease, x = no adipokine secreted.

**Table 1 nutrients-14-04944-t001:** Effects of Zea and Exc on body weight, weight gain, food intake, and relative visceral adipose tissue in HFD-induced obese rats.

Group	Control	HFD	HFD +Zea	HFD+ Exc	HFD + Zea/Exc
Initial body weight (g)	201.71 ± 6.58	203.21 ± 5.38	200.29 ± 6.59	202.67 ± 4.18	203.79 ± 5.22
Final body weight (g)	367.88 ± 7.22	494.76 ± 9.26 ^###^	488.79 ± 9.29	484.38 ± 14.94	484.38 ± 9.26
Weight gain (g)	166.17 ± 6.48	291.55 ± 11.77 ^###^	288.49 ±12.71	281.71 ± 13.16	280.6 ± 9.832
Food intake (g/day)	29.33 ± 0.40	24.48± 0.36 ^###^	24.19± 0.57	23.77 ± 0.71	23.5 ± 0.52
Energy intake (kcal/day)	113.33 ± 1.55	129.53 ± 1.03 ^###^	127.85 ± 2.10	125.71 ± 2.07	124.30 ± 1.66
Relative visceral adipose tissue (%)	1.45 ± 0.19	5.24 ± 0.46 ^###^	5.15.33 ± 0.48	4.86.33 ± 0.47	5.03 ± 0.13

Data are expressed as means ± SEM, *n* = 12. ^###^
*p*< 0.001 vs. control.

## Data Availability

Not applicable.

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
