# Peer review of "Evaluation of the Anti-Obesity Effect of Zeaxanthin and Exercise in HFD-Induced Obese Rats"

_nutrients, 2022, doi:10.3390/nu14234944_

Round 1

Reviewer 1 Report

The manuscript entitled “Evaluation of the Anti-Obesity Effect of Zeaxanthin and Exercise in HFD-Induced Obese Rats”, authored Mona Al-thepyani et al., deals with an evaluation of the Anti-Obesity Effect of Zeaxanthin and Exercise. The topic of this manuscript is important and current, and results could be interesting for readers. However, I miss the future perspective of using the results described in the manuscript. Please complete it before the work will be further processed.

Author Response

Dear professor 

Thank you for your time and for your valuable comments. 

Future prospective has been added to the conclusion section (Line 785-790)

Best regards 

Reviewer 2 Report

The manuscript by Al-thepyani and colleagues investigates the potential synergistic anti-obesity effect of carotenoid Zeaxantin and exercise in a diet-induced obesity model in rats. Authors showed that HFD significantly increased visceral adipose tissue, oxidative stress, and inflammation biomarkers and reduced insulin, high-density lipoprotein, and antioxidant parameters which goes along with the diet-induced obesity model. Treatments with Zeaxantin, exercise and Zeaxantin plus exercise show a trend toward reduction of body weight gain, and reduced tri-acylglycerol, glucose, total cholesterol and nitric oxide levels and significantly increased catalase and insulin compared to the HFD group. Overall, the experiments are well-conducted, and the data are novel and of interest. On the other hand, the authors tend to some extent overinterpret their results, and the manuscript is lacking discussion of some of the potential caveats and additional work is required to improve the quality of the manuscript.

Comments:

1.      Materials and methods section.  Section 2.4. Measurement of body weight, food intake, and relative weight of visceral adipose tissue - in a formula that the authors provided titled The ratio of fat to body weight should be changed to The ratio of visceral fat to body weight because the authors did not measure the amount of subcutaneous fat that in HFD mice can present a high percentage of overall fat content.  

Food intake – I am guessing that these rats were group-housed, so please clarify how the food measurement was performed and later on how was the effect the statistical analysis of energy intake

Section 2.5. Determination of serum biochemical parameters – Please clarify if these mice were fasted, which would be especially important in the context of blood glucose level measurements.

2.      Overall there was no effect in body weight reduction except a slight decrease in weight gain and visceral fat content in groups treated with Zea or doing daily exercise so we can’t be talking about an anti-obesity effect in the context of reduced weight gain. The authors should include error bars in Figure 2.

3.      It would strengthen the manuscript if the authors could provide visceral fat staining (taking into account that this fat was measured and presumable collected) and measure the adipocyte size.

4.      Swimming as an exercise model used in this study can be a potent stressor (as authors acknowledge in the discussion) but on the other side physical activity like swimming can decrease levels of corticosterone in unstressed rats (see PMID: 32937768). In this context, it would be important to check levels of CORT and ACTH after 5 weeks of exercise and see the effect of Zea on CORT and ACTH levels.

5.      Glucose levels in HFD were increased but insulin levels were decreased in HFD mice, which is contrary to the expected results (according to the authors) when it comes to insulin level. This means that in their diet-induced obesity model they did not see insulin resistance and the authors provide some possible explanations for this. To better understand what was going on authors should have to perform GTT and ITT. Taking into account that that would require a new experiment set up authors should acknowledge that lacking GTT and ITT in the interpretation of blood glucose data.

6.      I encourage the authors that present their data in bar graphs with individual values included.

Author Response

Dear Professor 

Thank you for your time and your constructive input. 

Please refer to our responses to your valuable comments in the attachment. 

Round 2

Reviewer 2 Report

Manuscript has been improved enough to be accepted and authors addressed all raised questions